# Bimetal–Organic Framework-Loaded PVA/Chitosan Composite Hydrogel with Interfacial Antibacterial and Adhesive Hemostatic Features for Wound Dressings

**DOI:** 10.3390/polym15224362

**Published:** 2023-11-09

**Authors:** Nan Zhang, Xiuwen Zhang, Yueyuan Zhu, Dong Wang, Ren Li, Shuangying Li, Ruizhi Meng, Zhihui Liu, Dan Chen

**Affiliations:** 1College of Environment and Safety Engineering, Qingdao University of Science and Technology, Qingdao 266042, China; 2Shandong Engineering Research Centre for Marine Environment Corrosion and Safety Protection, Qingdao University of Science and Technology, Qingdao 266042, China; 3Shandong Engineering Technology Research Centre for Advanced Coating, Qingdao University of Science and Technology, Qingdao 266042, China; 4Qingdao High-Tech Industry Promotion Centre (Qingdao Technology Market Service Centre), Qingdao 266112, China

**Keywords:** bimetal-MOF, antibacterial, hydrogel, chitosan, hemostatic

## Abstract

Silver-containing wound dressings have shown attractive advantages in the treatment of wound infection due to their excellent antibacterial activity. However, the introduction of silver ions or AgNPs directly into the wound can cause deposition in the body as particles. Here, with the aim of designing low-silver wound dressings, a bimetallic-MOF antibacterial material called AgCu@MOF was developed using 3, 5-pyridine dicarboxylic acid as the ligand and Ag^+^ and Cu^2+^ as metal ion sites. PCbM (PVA/chitosan/AgCu@MOF) hydrogel was successfully constructed in PVA/chitosan wound dressing loaded with AgCu@MOF. The active sites on the surface of AgCu@MOF increased the lipophilicity to bacteria and caused the bacterial membrane to undergo lipid peroxidation, which resulted in the strong bactericidal properties of AgCu@MOF, and the antimicrobial activity of the dressing PCbM was as high as 99.9%. The chelation of silver ions in AgCu@MOF with chitosan occupied the surface functional groups of chitosan and reduced the crosslinking density of chitosan. PCbM changes the hydrogel crosslinking network, thus improving the water retention and water permeability of PCbM hydrogel so that the hydrogel has the function of binding wet tissue. As a wound adhesive, PCbM hydrogel reduces the amount of wound bleeding and has good biocompatibility. PCbM hydrogel-treated mice achieved 96% wound recovery on day 14. The strong antibacterial, tissue adhesion, and hemostatic ability of PCbM make it a potential wound dressing.

## 1. Introduction

The infection of skin wounds can increase and lead to more treatment costs, treatment failure, and even death [1,2]. Wound healing is a multistage coordinated process that includes four phases: hemostasis, inflammation, proliferation, and remodeling [3,4]. Bacterial infection at any stage of wound recovery may turn an acute wound into a chronic wound or result in fibrillary degeneration [5]. The use of anti-infective wound dressing has become the main strategy in wound recovery maintenance [6]. Hydrogel wound dressings have been widely studied in recent years because of their multifunctionality [7]. According to the theory of wet wound healing, a moist environment facilitates wound recovery, and hydrogel trauma dressing can maintain a sterile and moist environment on the wound surface to accelerate wound healing [8,9]. Consequently, in order to accomplish this objective, we created a hydrogel wound dressing with MOFs that possess antibacterial properties.

Traditional hemostatic dressings are usually designed to be dry [10]. Hydrogels create many possibilities for creating dressings that are moist and highly viscous and are barriers to microbes [11,12,13]. Hydrogels are advantageous for wound healing due to their water permeability, ability to facilitate metabolite exchange, and mechanical strength, making them an ideal dressing for joint wounds [14,15]. Polyvinyl alcohol (PVA) is a synthetic polymer, and the freeze–thaw cycle of aqueous solutions of polyvinyl alcohol forms crystallites as crosslinking nodes [16,17]. In a previous preliminary study by Zhang M, PVACS and Ag@MOF were used to synthesize hydrogels, which had antibacterial properties. On this basis, we further developed bimetallic MOF hydrogels [18].

In practice, wound dressings are in direct contact with wounds, maintaining a sterile environment and good biocompatibility while maintaining wound moisture. Natural biomass material has the characteristics of degradability and good compatibility [19]. Biomaterials of high quality, such as collagen, starch, and CS, have been studied extensively in the field of biomedicine [20,21], with CS being derived from the deacetylation of chitin due to its strong adhesion and hemostatic and antibacterial properties. CS is one of the most promising wound dressing materials [22]. Utilizing grafting and adding modified materials, the antibacterial and hemostatic properties, biocompatibility, and other functions of CS-based dressings are expanded, which makes CS-based dressings more widely used [23].

Metal–organic frameworks (MOFs) are coordination polymers composed of organic linker ligands connected to metal ion nodes. An MOF has an extremely high surface area and abundant active sites that are widely used in gas capture, catalysis, medicine, and other fields [24]. The application of MOFs in medicine mainly depends on the activity of organic bridging ligands, metal ions, and surface sites. The remarkable permeability and consistent structure of MOFs have prompted widespread investigation in the biomedical domain. Two mechanisms by which MOFs inactivate bacteria include the interaction between the active site on the surface of the MOF and the bacterial surface, and metal ions leached from the MOF crystals disrupting bacterial cell membrane permeability. The large amount of three-dimensional spatial structure and surface area of MOFs provides many metal active sites on the surface, enhancing the lipophilicity of MOFs. The active site causes the peroxidation of lipids in the bacterial membrane, resulting in damage to the bacterial membrane.

We selected bioactive Ag^+^ and Cu^2+^ as metal nodes for the design of the MOF structure. The wound-healing process needs to promote the process of angiogenesis, collagen deposition, and wound re-epithelialization [25]. Copper-containing materials have been shown to have angiogenic, osteogenic, and antibacterial activities that promote wound healing [26,27]. Transition metal nanoparticles (NPs), especially Ag-NPs, have prominent effects on multidrug-resistant bacteria. However, the excessive leaching of metal ions from metal nanoparticles is harmful to normal cells and can lead to serious systemic toxic effects [28,29]. In this study, the Cu^2+^ release of the MOF at the lowest bactericidal concentration (MBC) was 0.032 mg L^−1^, which was much smaller than the IC50 value of human umbilical vein endothelial cells (HUVECs) (327.9 μM) [30]. Metal ions that exist in the coordination structure of MOFs are released slowly in the physiological environment [31], so MOFs can achieve the purpose of the slow release of ions to reduce the concentration of metal ions on the wound [32]. Relevant studies have shown that HKUST-1 (Cu-containing MOFs), which is doped with folic acid, has low cytotoxicity of slow-release Cu^2+^, enhances cell migration, promotes collagen deposition and re-epithelialization, and avoids cytotoxicity that is easily caused by the application of copper salts or copper oxide [33]. Based on previous studies on Ag@MOF, a bimetallic MOF doped with a copper metal joint was synthesized in this study, which made the dressing have good antibacterial activity. Therefore, adding AgCu@MOF to the upper wound dressing is an ideal strategy for promoting antibacterial properties and wound healing.

As a whole, a bimetallic-MOF-loaded hydrogel wound dressing with antibacterial and adhesive properties was developed. The slow-release property of MOFs is closely related to their structure, and the low concentration of metal ions oozing out reduces the toxicity of the human body. Due to the special structure of MOFs, the active site of metal ions makes them have excellent antibacterial properties. Antibacterial dressings can isolate external bacteria from contact with the wound surface for a short period of time, damaging the physical structure of bacteria. The synergistic action of ions and CS promotes platelet and red blood cell agglutination and demonstrates tissue adhesion, which can be used for rapid wound hemostasis. Dressings with a certain release of ions show better RBC compatibility and reduce the impact on RBC rupture. The design of bimetallic MOFs in the application of wound dressings shows multiple functions, which has inspired the wide application of bimetallic MOF materials in other fields.

## 2. Experimental Section

### 2.1. Materials

Chitosan (degree of deacetylation ≥ 95%), polyvinyl alcohol 1799 (alcoholics degree 98.0~99.0%), Cu(NO_3_)_2_·3H_2_O, AgNO_3,_ and 3,5-pyridinedicarboxylic acid (H_2_PYDC) were obtained from Hushi (Sinopharm Chemical Reagent Co., Ltd., Shanghai, China).

### 2.2. Synthesis of MOFs

A mixture of AgNO_3_ (0.32 g, 1.88 mmol) dissolved in deionized water (10 mL) and H_2_PYDC (0.1 g, 0.6 mmol) dispersed in water (10 mL) through ultrasonication was sealed in a Teflon-lined autoclave and heated at 120 °C for 24 h. The resultant colorless acicular crystal was parted by centrifuging (at 4800 rpm, 10 min), washed thrice with water, and dried at 50 °C for 5 h [34].

A mixture of Cu(NO_3_)_2_·3H_2_O (0.445 g, 1.83 mmol) dissolved in deionized water (10 mL) and H_2_PYDC (0.14 g, 0.92 mmol) dispersed in water (10 mL) through ultrasonication were sealed in a Teflon-lined autoclave and heated at 120 °C for 24 h. After dumping the supernatant following precipitation, the resultant blue acicular crystal was washed several times with water and dried at 50 °C for 5 h.

AgNO_3_ (0.15 g, 0.94 mmol) and Cu(NO_3_)_2_·3H_2_O (0.222 g, 0.92 mmol) were dissolved in 10 mL of deionized water, and H_2_PYDC (0.14 g, 0.92 mmol) was dispersed by ultrasonication in 10 mL of deionized water until completely dispersed. The two liquids were mixed, transferred into a Teflon autoclave reactor and heated at 120 °C for 24 h. A light-blue precipitate was obtained by centrifugation, and the precipitate was washed with deionized water and dried at 50 °C for 5 h. The precipitate was then extracted and dried at 50 °C for 1 h.

### 2.3. Preparation of Hydrogels Containing MOFs

We dissolved PVA in water while stirring at 80 °C until completely dissolved to obtain a 10 wt% PVA solution. MOFs were added to the PVA solution so that the content of MOFs in the final hydrogel was 1 mg g^−1^, which was synthesized in nitrogen protection. We dissolved CS completely in water and added acetic acid to form a sticky 0.1 M concentration acetic acid solution. This was followed by stirring at 60 °C for 0.5 h to obtain a 2 wt% CS solution. Both solutions were balanced and mixed to acquire PVACS (PC) hydrogel, PVACS/Cu@MOF (PCCuM) hydrogel, PVACS/Ag@MOF (PCAgM) hydrogel, and PVACS/AgCu@MOF (PCbM) hydrogel. The solution was cast on a plastic mold and was left to vacuum-degas. The samples were frozen four times at 20 °C to form hydrogels.

### 2.4. Characterization

The microstructure of the samples was characterized with a JSM-6700F scanning electron microscope (SEM) (JEOL, Tokyo, Japan) with an accelerating voltage of 10.00 kV. The structure and phase composition of the samples were characterized with an X-ray diffractometer (ULTIMALV, Japan Science Corporation, Tokyo, Japan) operating at a voltage of 10 kV, a current of 20 mA, and a scanning angle range of 5~60°. The changes in the functional groups of the materials were characterized with an IRAffinity-1 spectrometer (Shimadzu, Kyoto, Japan) from 400 to 3600 cm^−1^ at a scanning speed of 2 cm^−1^. The mass changes of the samples were observed with an SDT-Q600 thermal analyzer (TA instruments, New Castle, DE, USA) under nitrogen protection in the range of 50 to 700 °C with a ramp rate of 10 °C/min. The content of elemental silver and copper in the hydrogels was determined using an inductively coupled plasma emission spectrometer (ICP-OES, Avio200, PerkinElmer, Waltham, MA, USA).

### 2.5. Water Contact Angle

A high-speed camera was used to measure the water contact angle of the sample, and a probe was used to drop about 3 μL of ultra-pure water onto the surface of a clean hydrogel dressing [35]. Software was used to calculate the water contact angle, and three groups of repeated experiments were used to obtain the average value.

### 2.6. Swelling Behavior

The swelling rate of the hydrogel was determined using the weight method [36]. The wet hydrogels were immersed in phosphate-buffered saline (PBS) (0.01 M, pH 7.4) at 37 °C. The excess liquid on the surface of the filter paper was aspirated at intervals and then weighed until the sample mass reached equilibrium. The swelling rate was calculated as follows [37]:(1)$$Weight\;remaining\left(\%\right)=\frac{W_{a}-W_{b}}{W_{b}}\times 100\%$$
$W_{a}$ and $W_{b}$ are the weights of the hydrogels before and after immersion in PBS.

### 2.7. Water Retention Studies

The hydrogel weight was balanced in PBS buffer, and the samples were placed in the oven at 37 °C. The samples were removed and weighed at regular intervals until the sample mass no longer decreased. Water retention was calculated as follows [38]:(2)$$WR\left(\%\right)=\frac{W_{t}}{W_{e}}\times 100\%$$
$W_{t}$ is the weight of the hydrogel at time t, $W_{e}$ is the weight of the hydrogel after equilibrium.

### 2.8. Water Vapor Permeability (WVP)

A 35 mm diameter hydrogel dressing was fixed in the mouth of a 25 mm diameter beaker container containing 10 mL of water and placed in an incubator at 37 °C and 35% relative humidity, and the system weight was weighed at certain time intervals [39]. The water vapor transmission rate (WVTR) was calculated by the following equation:(3)$$WVTR\left(\%\right)=\frac{{WV}_{loss}}{A_{t}}\times 100\%$$
$WV_{loss}$ reduction in system weight after t time, and A is the test area of the sample in m^2^. Each experiment was repeated three times to take the average value.

### 2.9. Metal Ion Release

AgCu@MOF and PBS buffer were used to prepare the AgCu@MOF suspension, and the concentration gradients were 80 ppm, 40 ppm, 20 ppm, 10 ppm, and 5 ppm for the experimental group. The concentrations of copper and silver ions were measured spectrophotometrically after shaking at 120 rpm for 24 h at 37 °C. Methods for determining metal ion concentrations are described in the Appendix A [40,41]. We set up three parallel groups.

### 2.10. Antibacterial Activity Assay In Vitro

The antibacterial activity of AgCu@MOF in vitro was evaluated by the plate count method [42]. *Escherichia coli* (ATCC25922) and *Staphylococcus aureus* (CMCC(B)26003) were selected to evaluate the antibacterial performance in vitro. The bacteria were cultured in LB broth and oscillated at 37 °C and 120 rpm for 12–16 h. The bacterial suspension was diluted to 10^7^ CFU/mL with LB broth and divided into four groups, and the concentration of MOFs was made up to 100ppm by adding (1) PBS, (2) Cu@MOF, (3) Ag@MOF and (4) AgCu@MOF, respectively. The bacterial solution was incubated at 37 °C and 120 rpm for 12 h. After 12 h, 100 µL of each bacterial solution was taken and used to coat a solid medium for overnight incubation at 37 °C. Image J (version: 1.52i) was used to calculate the number of colony-forming units (CFUs), and the antibacterial efficiency was calculated according to the following formula:(4)$$Percentage\;Survival\;rate\left(\%\right)=\frac{Number_{samples}}{Number_{control}}\times 100\%$$

As for exploring the minimum bactericidal concentration (MBC) of AgCu@MOF, bacteria were collected by centrifuge collection of the bacterial suspension and diluted to 10^6^ CFU/mL with PBS; then, AgCu@MOFs of different qualities were added, and the samples were cultured at 37 °C and 120 rpm for 12 h [43]. After incubation for 12 h, 100 µL of bacterial suspension was removed and used to coat an LB solid medium, and it was incubated at 37 °C overnight. Image J was used to calculate the number of colonies on the plate. Five experimental groups with AgCu@MOF concentrations of 5 ppm, 10 ppm, 20 ppm, 40 ppm and 80 ppm were established.

To explore the antibacterial properties of hydrogels, sterilized hydrogels of PC, PCAgM, PCCuM and PCbM were incubated with the bacterial suspension at 37 °C and 120 rpm for 12 h [44]. After incubation for 12 h, 100 µL of bacterial suspension was used to coat the LB solid medium. It was incubated at 37 °C overnight, and the number of colonies on the plate was calculated by Image J.

To explore the bactericidal effect of PCbM, the bacterial solution was diluted to 10^5^ CFU/mL with PBS, and then cultured with PCbM at 37 °C and 120 rpm [45]. Sample every 20 min and dilute 10^1^, 10^2^, 10^3^ times. 40 µL diluted bacterial solution was added into LB solid medium and incubated at 37 °C overnight. Image J was used to calculate the number of colonies on the plate. At the same time, the antibacterial activity of hydrogels was demonstrated by the disk diffusion method. Then, 100 µL of 10^8^ CFU/mL bacterial suspension was applied to LB solid medium, affixed with disinfected discs of different hydrogels, and incubated overnight at 37 °C. After incubation, the size of the inhibition ring was recorded.

The vigorously growing bacteria in the medium and AgCu@MOF-treated bacteria were collected separately. Bacteria were mixed with sterile PBS solution and centrifuged at 8000 rpm for 3–5 min, and the supernatant was discarded. To the precipitate containing bacteria, 2.5% glutaraldehyde fixative was added and refrigerated at 4 °C for 2 h. The well-fixed bacteria were washed three times using PBS buffer by centrifugation under the same conditions. Then, they were washed once each with concentrations of 30/50/70/80/90/95/100%, respectively. The purpose of this operation was to gradually displace the water in the bacterial samples. Finally, the samples that were completely dehydrated were dropped on a silicon wafer and dried naturally before they could be observed using SEM.

### 2.11. In-Vitro Blood Clotting Test

We took 20 × 20 mm samples and placed them in a test tube, incubated at 37 °C for 5 min, added 1 mL of rabbit anticoagulation, incubated at 37 °C for 3 min, added 40 μL of CaCl_2_ solution (concentration 0.2 mol/L) and started timing [46]. Starting with the addition of CaCl_2_, we tilted the tube to 45° every 15 s to see if the blood was clotting. A blank control group was established. Each group of experiments was repeated three times.

### 2.12. In-Vitro Haemolysis Test

We used the hemolysis activity assay to evaluate the hemocompatibility of the hydrogel [47]. Defibrinated sheep blood was mixed with 0.9% saline at a ratio of 1:1.25 to obtain diluted whole blood. Then, 1 × 1 cm samples were washed three times with 0.9% saline and immersed in 10 mL of saline. The immersion was repeated two times and 0.2 mL of diluted whole blood was added to the system and then incubated at 37 °C for 1 h. The absorbance of the supernatant after centrifugation was measured at 545 nm. A positive control group with 10 mL of distilled water and a negative control group with 10 mL of saline was established. Each group of experiments was repeated three times.

### 2.13. Adhesion of Red Blood Cells (RBCs) and Platelets

We incubated samples of size 10 × 10 mm in defiber sheep blood for 30 min at 37 °C, rinsed three times with PBS to remove unadhered and loose red blood cells, and fixed them in 2.5% glutaraldehyde solution for 3 h [18]. The erythrocytes were dehydrated in an ethanol solution with a concentration gradient of 30/50/70/80/90/95/100%. The surface erythrocyte morphology of the samples was observed by SEM. The platelet-rich plasma clear solution was obtained by the centrifugation of potassium oxalate anticoagulated rabbit blood at 1000 rpm for 10 min. The platelet adhesion assay was performed in the same way as the erythrocyte adhesion assay.

### 2.14. In-Vivo Hemostatic Properties of PCbM in Mouse Tail Amputation and Liver Injury Models

Twelve 8-week-old male BALB/c mice were selected and divided into four groups [48]. The mice were anesthetized with isoflurane before the experiment. PCbM hydrogel was applied to the bleeding model of mouse tail amputation and liver incision to verify the tissue adhesion and hemostasis of PCbM under different conditions. In the tail amputation model, the tail was cut off 3 cm from the tail tip. In the liver bleeding model, a 5 mm wound was cut on the mouse liver after liver exposure. PCbM hydrogel was applied immediately after exposure to the wound, and the bleeding area was photographed at different times. The amount of bleeding on the filter paper after each bleeding test was weighed, and the time to stop the bleeding was recorded. A blank control group was set.

### 2.15. Cell Viability Assay

The cytotoxicity of PC, PCAgM, PCCuM and PCbM hydrogels was studied by the CCK-8 method [49]. Mouse L929 cells were inoculated at a density of 1 × 10^−5^ cells per well on 96-well plates with Dulbecco’s modified Eagle medium (DMEM) containing 10% fetal bovine serum. The cells were incubated overnight in a 37 °C incubator at a CO_2_ concentration of 5%. The following morning, the stale medium was discarded, and fresh DMEM medium with 10% fetal bovine serum was added with the aim of ensuring cell viability. Then, 5 mg of UV-sterilized PC, PCAgM, PCCuM and PCbM hydrogels were added to each cell well plate. The cells were then incubated at 37 °C for 24 h in an incubator with a CO_2_ concentration of 5%. At the end of the incubation, we aspirated and discarded the medium, and 10 µL of CCK-8 was added to each well and incubated for 3 h. The optical density (*OD*) values of the mixtures were detected at 450 nm using an enzyme marker (Multiskan FC, Thermo Fisher, Waltham, MA, USA). Cell viability was calculated by the following formula:$$Cell\;viability\;\left(\%\right)=\frac{OD_{samples}-OD_{blank}}{OD_{control}-OD_{blank}}\times 100\%$$

### 2.16. Confocal Laser Scanning Microscopy

To assess the damage of AgCu@MOF in PCbM to the cell membranes of *E. coli* and *S. aureus*, the samples were analyzed by double fluorescence staining with Syto^®^9 and propidium iodide (PI) using the method of CLSM [50]. Syto^®^9 stains the DNA of live bacteria, which gives a green fluorescence. The PI dye cannot pass through the intact cell membrane, which is permeable in dead cells, so PI can stain dead cells, giving them a red fluorescence. *E. coli* and *S. aureus* were treated with 1 × MIC concentration of AgCu@MOF for 1 h, and then the samples were stained with 2 μM of Syto^®^9 and 30 μM of PI in the dark for 15 min, washed with PBS and centrifuged, and observed by CLSM at 488 nm and 561 nm. The control group was not subjected to antimicrobial treatment.

### 2.17. In-Vivo Wound Healing in a Full-Thickness Skin-Defect Model

Sixteen 8-week-old BALB/c mice with an average weight of 35 g were divided into four groups for wound-healing experiments [51]. The mice were anesthetized using isoflurane, and a circular, 10-mm diameter, full-layer skin wound was created on each mouse’s back to evaluate the PCbM hydrogel’s wound-healing effect. Different groups of mice were treated with Tegaderm film (3M, St. Paul, MN, USA), PCAgM, PCCuM, and PCbM hydrogels for the wounds, where the Tegaderm film group was used as the control group. The dressings were changed daily, and the wound area was measured and photographed on days 3, 7, and 14, respectively. The wound-healing process was assessed by the wound area, and the healing rate (*HR*) was calculated as follows:(5)$$HR=\frac{S_{0}-S}{S_{0}}\times 100\%$$
where $S_{0}$ is the size of the trauma area on the day the wound was created and S is the size of the trauma area on that day.

Skin tissue specimens were taken on days 3, 7 and 14, respectively, and then treated with 4% paraformaldehyde. Pathological sections were stained with hematoxylin-eosin (H&E) with a light microscope (ECLIPSE CI-L, Nikon, Tokyo, Japan) and analyzed pathologically.

### 2.18. Statistical Analysis

All the data were expressed as means ± standard deviations. Comparisons among the three groups were analyzed with one-way ANOVA using SPSS (version 22.0, SPSS Inc., Chicago, IL, USA).

## 3. Results and Discussion

### 3.1. Characterization of MOFs and Hydrogels

The FTIR analysis (Figure 1a) demonstrates the occurrence of interaction between CS and AgCu@MOF. For PC, characteristic peaks were observed at 1380 cm^−1^ (bending vibration in the -OH plane), 1090 cm^−1^ (stretching vibration peak of C-O bond in hydroxyl group), and 2923 cm^−1^ (vibration peak of O-H on -COOH). The weak characteristic peaks of pyridine were observed at 1266 cm^−1^ (Stretching vibration peak of C-H on pyridine ring) and 1604 cm^−1^ (C=C, C=N double stretching vibration peak). Compared with PC, the FTIR of PCAgM, PCCuM and PCbM was enhanced at the peak in the range of 1680~1575 cm^−1^, and the bending vibration peak was at 1300~1200 cm^−1^. It was speculated that the effect of AgCu@MOF on CS enhanced the activity of -COOH. CS showed a dispersive broad peak around 21° because of the amorphous nature (Figure 1b). The figure shows the XRD patterns for PCAgM, PCCuM and PCbM and the obvious crystalline peaks around 11.7°, 15.6°, 25.9° and 35.6°. The XRD pattern of the hydrogel with AgCu@MOF showed obvious peaks, indicating that AgCu@MOF could be present in the hydrogel without visible structural collapse. The TGA curves show PC, PCAgM, PCCuM and PCbM (Figure 1c). Only one mass-loss phase occurred in PC after 200 °C; the remaining mass of the residual amounts is about 16%. Two mass-loss phases existed in PCAgM, PCCuM, and PCbM. The first mass-loss step occurs at 280 °C, and the water molecules bound by hydrophilic groups in the sample are removed. The second weightlessness phase begins with the decomposition of hydroxyl, carboxyl and pyridine rings at 323 °C, and then all the remaining mass of the residual amounts is about 31%. The addition of AgCu@MOF leads to an increase in decomposition temperature and residual amount, which improves the thermal stability of the dressing. After AgCu@MOF was soaked in PBS buffer for 24 h, the leaching amounts of Ag^+^ and Cu^2+^ were measured. The concentrations of Ag^+^ and Cu^2+^ were positively correlated with the concentrations of AgCu@MOF and decreased with the decrease in AgCu@MOF concentration (Figure 1f). The metal ions in the AgCu@MOF cause a strong electrostatic interaction between the metal joints and the carboxylate joints, which gives the frame excellent stability. The special three-dimensional spatial structure reduces the leaching rate of metal ions. Taking the data in the figure as an example, the maximum leaching amount of Cu ions is only 0.080%, while the maximum leaching amount of silver ions is 0.048%.

### 3.2. Micromorphology of MOFs and Hydrogels

The morphologies of Ag@MOF, Cu@MOF, and AgCu@MOF were observed by SEM (Figure 2a–c), and all the MOFs showed a rod structure. The images show a wide size distribution of MOFs, ranging from 1 to 50 um in width and 10 to 500 um in length (Appendix A). An energy dispersion spectrometer (EDS) confirmed the uniform distribution of Ag and Cu on the surface of AgCu@MOF. The topography of the hydrogel was analyzed by SEM. PVA generates a hydrogen bond crosslinking network in the freeze–thaw cycle, forming a connected dense porous gel. The diameter of the hole in the PC is fine (Figure 2e), and the average diameter is 3~4 μm, but in PCBM (Figure 2i), the holes with a greater diameter of 10 μm are seen in the PC. The addition of Ag@MOF and AgCu@MOF resulted in larger pore sizes for PCAgM and PCbM than for PC and PCCuM hydrogels. Since the amino (-NH_2_) and hydroxyl (-OH) groups on chitosan are the main reactive groups with silver ions, the chitosan chains bind to Ag^+^ through chelation reactions [52]. The addition of AgCu@MOF increased the pore size of the hydrogel network and changed the crosslinking density of the PVACS hydrogel network. The large pore structure of the hydrogel is conducive to the absorption of liquid seepage on the wound surface. The elastic network of hydrogels provides space for water filling and loss, which provides the possibility for it to be used as a wound dressing [53]. The element mapping reflects the uniform distribution of Ag and Cu on PCbM, indicating that Ag and CS have been mixed evenly (Appendix A).

### 3.3. Physical Properties of Hydrogels

The dressing was soaked in PBS to simulate the situation in which the dressing absorbs tissue exudate on the wound (Figure 1d). The water absorption and swelling of PCAgM and PCbM are good, which is consistent with the predicted conclusion on structural morphology. Hydrogels can absorb more simulated body fluids. Higher swelling properties help the dressing absorb blood and control bleeding. With the increase in Cu@MOF concentration, the water retention rates of PC, PCAgM, PCCuM and PCbM in 10 h were 4.9%, 5.1%, 4.3% and 2.6%, respectively (Figure 1d). The reduced crosslinking density of PCbM hydrogels promotes the release of water vapor from the hydrogel surface and reduces wound water retention.

The water contact angle of the samples was 18° ± 3° (Figure 2g), indicating that the samples had good hydrophilicity. The water vapor transmission rates of PC, PCAgM, PCCuM and PCbM are 2170, 2180, 2504 and 2608 g m^−2^ over 24 h^−1^, respectively (Figure 2f). Similarly, WVTR indicates that the addition of Ag@MOF and AgCu@MOF can properly improve the crosslinking network of the hydrogel, and the effect of this on water retention rates and swelling behavior increases the water absorption capacity of the hydrogel and can quickly absorb body fluid from the wound. The WVTR of normal skin is 200~300 g m^−2^ in 24 h^−1^, and the WVTR of open-skin wounds is 10 times that of normal skin, so the appropriate WVTR of wound dressings should be between 2000 and 3000 g m^−2^ in 24 h^−1^ to maintain the dynamic balance of wound water content [54]. Proper water vapor permeability prevents the hydrogel from dehydrating the skin surface and sticking to the wound dressing.

In order to understand the content of metal ions in the hydrogel, we determined the content of metal ions in the hydrogel using ICP-OES measurement [55]. The results were 0.534 ± 0.011 mg/g hydrogel for silver concentration in PCAgM and 0.376 ± 0.006 mg/g hydrogel for copper concentration in PCCuM. The silver and copper concentrations in PCbM were 0.322 ± 0.006 and 0.039 ± 0.003 mg/g hydrogel, respectively. As indicated in the EDS elemental mapping, the introduction of Cu sites into Ag@MOF resulted in lower Cu content in the bimetallic MOF. The lower Cu content in the bimetallic MOF results in a low Cu concentration in the PCbM hydrogel.

### 3.4. Antibacterial Activity of the PCbM Hydrogels

Wound dressings should have appropriate antimicrobial properties, which can reduce pathogen formation in the wound [14]. We discussed the antimicrobial properties of MOFs against *E. coli* and *Staphylococcus aureus* (Figure 3a). The survival of different MOFs incubated with bacteria at the same concentration was compared. The results showed that the antibacterial activity of AgCu@MOF was higher than that of an MOF composed of a single metal ion ligand. Cu@MOF showed no antibacterial activity, and Ag@MOF showed only 95% antibacterial activity against *Escherichia coli* (Figure 3e). However, AgCu@MOF with two metal ion sites showed significant antibacterial activity, with a bacterial survival rate of 0%. Furthermore, the bacteriostatic effect of AgCu@MOF was concentration-dependent. The survival rate of *E. coli* and *S. aureus* treated with 20 ppm AgCu@MOF decreased significantly. The survival rate of *E. coli* treated with 40 ppm AgCu@MOF decreased to 0%. This indicates that the minimum bactericidal concentration of AgCu@MOF for *E. coli* is 20 ppm (Figure 3b,f). The results showed that AgCu@MOF had a good antibacterial effect. The antibacterial activity of hydrogels containing MOFs was studied (Figure 3c). In addition, the bactericidal rate of PCbM hydrogels containing 1000 ppm and 500 ppm AgCu@MOF was compared against bacteria when incubated with bacteria. The bactericidal rate of the low-AgCu@MOF PCbM hydrogel against *Escherichia coli* and *Staphylococcus aureus* reached 70% and 7%, respectively, while the bactericidal efficiency of the high-AgCu@MOF PCbM reached 100% (Figure 3g). Based on studying the bactericidal performance of the PCbM hydrogel, the bactericidal performance was further discussed; that is, the influence of the incubation time of bacteria with PCbM on the survival rate of bacteria was explored. The efficiency of PCbM hydrogel increased with time, killing more than 96% of *E. coli* and 100% of *Staphylococcus aureus* within one hour (Figure 3d,h). Finally, the disk diffusion method was used to verify the antibacterial properties of PC, PCCuM, PCAgM and PCbM hydrogels, and the experimental results showed that the antibacterial zone size of the hydrogels was increasing (Appendix A). PCAgM and PCbM had large inhibition zones, and the inhibition zones of PCbM against *Escherichia coli* and *Staphylococcus aureus* were 8.79 mm and 8.37 mm, respectively (Appendix A).

PCbM was immersed in the bacterial solution, and we observed the attachment of bacteria on its surface, and the results proved that PCbM could prevent bacteria from attaching and growing on its surface. So, there are no bacteria attached to the surface of the hydrogel, and they cannot penetrate it. SEM images showed that no bacteria attached to the surface could be observed, indicating that PCbM could prevent the formation of biofilms (Appendix A). The effect of PCbM gel on the morphology of bacteria was studied using a scanning electron microscope (SEM). The cell wall of normal *Escherichia coli* and *Staphylococcus aureus* was complete and non-collapsing (Figure 3i), and their morphologies were smooth rod and ball, respectively. The cell wall surface of PCbM hydrogel-treated bacteria contracted severely, and the bacteria killed by AgCu@MOF collapsed and adhered to the surface of AgCu@MOF in large numbers. The results indicate that PCbM can kill bacteria by destroying the cell walls of bacteria. In particular, we investigated the release of metal ions in AgCu@MOF and bacterial cultures at minimum inhibitory concentrations, thereby reflecting the actual concentration of metal ions in the antibacterial system. When AgCu@MOF concentration is 20 ppm, the contents of Cu^2+^ and Ag^+^ ions in bacterial suspension are 0.013 ppm and 0.008 ppm, and when AgCu@MOF concentration is 40 ppm, the corresponding contents of Cu^2+^ and Ag^+^ are 0.032 ppm and 0.016 ppm (Figure 1f). This concentration is lower than in previous studies, indicating that the MBC values of Cu^2+^ and Ag^+^ are about 10^−5^~10^−4^ M and 2.5 × 10^−7^~10^−6^ M (6.35~63.54 ppm and 0.027~0.107 ppm), respectively. These results indicate that the antibacterial effect of AgCu@MOF is not simply ion rerelease but the combined action of surface-active sites and metal ion release. The active-surface metal sites on MOF crystals could easily oxidize bacterial membranes or change the transmembrane potential [56]. The metal site on the surface of AgCu@MOF interacts with the bacterial cell wall, which synergies with the release of metal ions to improve the permeability of the cell wall, leading to cell rupture, cell content flow out, and eventually bacterial death.

### 3.5. Confocal Laser Scanning Microscopy

To further illustrate the disruption of the cell membranes of *E. coli* and *S. aureus* noted in the SEM, the CLSM method was used to demonstrate the disruption of the permeability of the bacterial cell membrane. Since the cell membranes of live cells are structurally intact and will only be stained by Syto^®^9 to fluoresce in green, live *E. coli* and live *S. aureus* show bright-green fluorescence, as shown in Figure 4a,e. The cell membranes of live *E. coli* and live *S. aureus* are also shown to fluoresce in green. The dead bacteria in Figure 4d,h show mostly red fluorescence for *E. coli* and fragmented yellow fluorescence for *S. aureus* because of cell membrane damage. Analyses by CLSM showed that the three-site spatial structure and surface metal active sites of AgCu@MOF interacted with bacterial cell membranes and triggered cell membrane damage in *Escherichia coli* and *Staphylococcus aureus*.

### 3.6. Biocompatibility of Hydrogels

Surgical trauma or bleeding can be dangerous, and the ideal wound dressing should have a strong ability to stop bleeding. The hemostatic process involves the role of the vascular endothelium, platelets, coagulation pathways and the complex relationship with fibrinolysis [57]. Further, in-vitro coagulation time (CBT) was used to evaluate the coagulability of PCbM. The CBT test shows that PCbM hydrogel promotes blood clotting. Figure 4a shows that in the blank group, the CBT of PC, PCAgM, PCCuM and PCbM was 206, 252, 262, and 279 s, respectively. It is speculated that cationic CS is the cause of promoting blood coagulation. Free amino groups interact with negatively charged red blood cells and precipitate them, forming clots at the site of injury and forming red blood cells and platelet aggregates on the surface of the hydrogel. SEM images showed that a large number of erythrocytes and platelets were attached to the PC hydrogel surface (Figure 4e,f).

Good biocompatibility is a necessary condition for wound dressings. Hemolysis is the process by which the cell membrane of the red blood cell breaks, resulting in the release of oxygenated hemoglobin. The ideal dressing should have a low hemolysis rate. The biocompatibility of hydrogels was evaluated by hemolysis rate in vitro. The hemolysis rates of PC, PCAgM, PCCuM and PCbM were 7.9%, 3.6%, 14.3% and 9.4%, respectively (Figure 4a). The hydrogel prepared by us has a low hemolysis rate, which provides the possibility for the safe use of PCbM in wounds. SEM showed that after hemolysis, the cell shape was round and the surface of the cell membrane was destroyed, while the red blood cells treated with PCbM showed a normal double-sided concave cake shape (Figure 4b,c).

The cell survival rates of PC, PCAgM, PCCuM and PCbM were 98%, 96%, 98% and 97%, respectively (Figure 4d). PCCuM and PCbM had slightly higher cell survival rates than other hydrogels. The use of Cu^2+^ can facilitate wound healing by regulating a variety of growth factors to stimulate keratinocyte and fibroblast migration and the process of angiogenesis and collagen deposition [33]. Therefore, PCbM hydrogels have the potential to be used as dressings to promote wound healing.

### 3.7. In-Vivo Hemostatic Properties of PCbM in Mouse Tail Amputation and Liver Injury Models

Figure 4a shows PCbM’s ability to stop the liver from bleeding. When the liver was cut open, blood flowed out. PCbM hydrogel was applied to the wound, and blood flow was found to decrease immediately and there was no excessive bleeding. In the control group, the blood loss was 591 mg, and the bleeding time was 5.4 min (Figure 5b). In the PCbM group, the blood loss was 119 mg, and the bleeding time was 3.5 min. Liver blood loss was reduced by 20.1% compared with the control group. When the hydrogel was removed from the liver, the blood rushed out quickly. The tissue stickiness of the hydrogel seals the wound and reduces blood flow from the wound. Cutting off the tail exposes bone, connective tissue, skin and three large blood vessels [58]. Five seconds after the tail was cut, the hydrogel was applied to the wound, and we saw a decrease in bleeding (Figure 5c). The pictures show the bleeding within three minutes. In the control group, the blood loss was 225 mg, and the bleeding time was 5.9 min (Figure 5d). In the experimental group, the blood loss was 16 mg, and the bleeding time was 4.3 min. The blood loss in the experimental group was significantly less than that in the control group. It is speculated that the low crosslinking density of PCbM hydrogels can improve the adhesion of hydrogels to wet tissues by changing the water absorption of hydrogels. The wound adhesion of PCbM has the effect of reducing blood loss, so PCbM is expected to be used as a wound hemostatic dressing.

### 3.8. Wound-Healing Effect of PCbM Hydrogel in Mouse Full-Thickness Skin-Defect Model

PCbM hydrogel was further evaluated as a potential wound dressing using a BALB/c mouse whole-skin-defect model. Photographs and measurements of the wounds using different dressings at days 0, 3, 7, and 14 are shown in Figure 6a,c, respectively. The results showed that wound recovery was accelerated to varying degrees in the groups that used the hydrogel compared to the control group. The PCbM hydrogel group significantly increased the rate of wound healing, achieving 79% healing on day 7 and almost complete healing on day 14. The quantitative analysis of the wound-healing rate is shown in Figure 6d, and the speed of the degree of reduction of the wound area at the same time point was as follows: PCbM > PCCuM > PCAgM > control. In particular, PCbM showed more wound recovery areas at day 14 compared to any group, with a healing rate of 96%, which visualized that the epidermal layer had been formed in its entirety. The faster healing in the PCbM hydrogel group could be attributed to the pro-angiogenic effect of the high antimicrobial properties and the release of a small amount of Cu ions, which, due to the introduction of a small quantity of Cu ions in the MOF, modulated angiogenesis and further promoted wound healing. In addition, the extracellular matrix-like environment provided by the hydrogel for the wound facilitates cell proliferation and migration.

To further evaluate the effect of PCbM hydrogel on the wound-healing process, histological analysis of the skin tissue was performed by H&E staining on days 3, 7, and 14 after creating the wound, respectively. As shown in Figure 7g, the epithelial thickness was thicker in the PCbM hydrogel group compared to the control group, with the thickness of the epidermal layer reaching approximately 19 μm on day 14. Although the control group also formed an epithelial and dermal layer, there was still localized hemorrhage in the dermal layer, and localized clusters of blood cells were visible. Fibroblast proliferation on the dermis was seen with disordered collagen fiber arrangement, localized edema and laxity of collagen fibers, and the presence of inflammatory cell infiltration. The epithelium and connective tissue in the PCbM group were regular with high fibroblast density, and neoangiogenesis was seen with fewer neutrophils.

## 4. Conclusions

In this study, an antibacterial PCbM wound dressing with good biocompatibility was successfully synthesized with PVA, CS and AgCu@MOF. Based on the interaction between the active site of AgCu@MOF and the bacterial surface, the bacterial cell membrane is damaged. PCbM hydrogel has excellent antibacterial activity. PCbM kills the bacteria on the wound and inhibits the reproduction of common bacteria, making it an ideal antibacterial dressing. The presence of AgCu@MOF reduces the crosslinking density of the CS hydrogel dressing, which gives the hydrogel good water absorption properties and a high WVTR. It can absorb the tissue fluid from the wound and quickly discharge it, preventing wound fluid retention and creating a moist environment for the wound. PCbM has good biocompatibility and a low hemolysis rate, and the hydrogel has low cytotoxicity, introducing the possibility of using it as a new wound dressing. Therefore, PCbM hydrogel could become a multifunctional antibacterial and healing dressing, and it has the potential for further development.

## Figures and Tables

**Figure 1 polymers-15-04362-f001:**
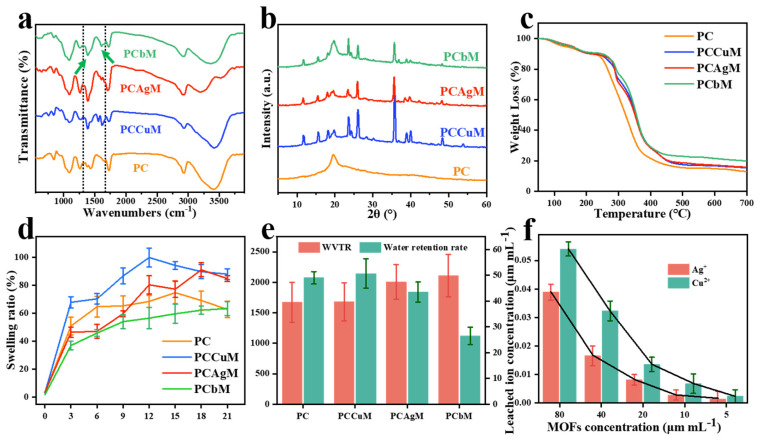
(**a**) FT-IR spectra, (**b**) XRD patterns and (**c**) TGA curves of PC, PCAgM, PCCuM and PCbM hydrogels. (**d**) Swelling rates of PC, PCAgM, PCCuM and PCbM hydrogels in PBS. (**e**) Water retention rate and WVTR of PC, PCAgM, PCCuM and PCbM hydrogels. (**f**) Total metal ions leached from AgCu@MOF solutions with concentrations ranging from 80 μg mL^−1^ to 5 μg mL^−1^.

**Figure 2 polymers-15-04362-f002:**
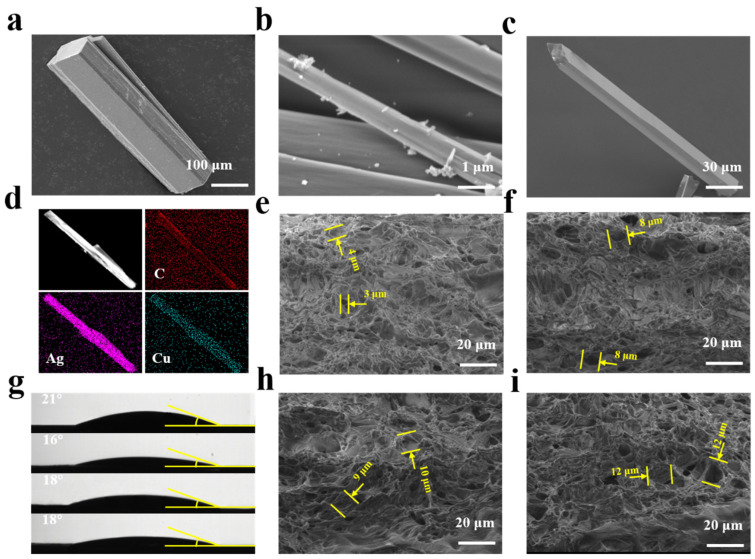
SEM images of (**a**) AgCu@MOF, (**b**) Ag@MOF, (**c**) Cu@MOF. (**d**) Elemental mappings of the C, Ag, and Cu of AgCu@MOF. Cross-section SEM image of (**e**) PC, (**f**) PCAgM, (**h**) PCCuM and (**i**) PCbM hydrogels after lyophilization. (**g**) Water contact angle of PC, PCAgM, PCCuM and PCbM hydrogels.

**Figure 3 polymers-15-04362-f003:**
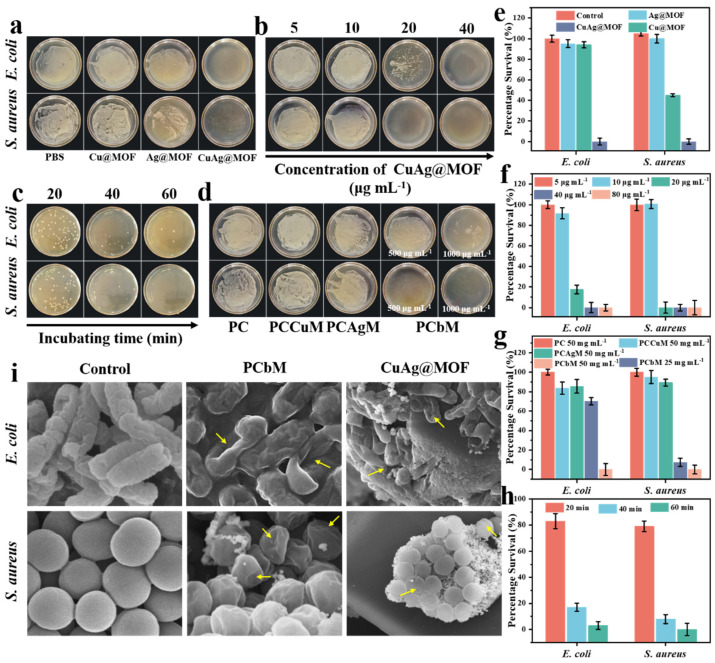
Colony growth of *Escherichia coli* and *Staphylococcus aureus* was performed using the plate counting method. (**a**) Bacterial samples were treated with (1) PBS, (2) Cu@MOF, (3) Ag@MOF and (4) AgCu@MOF. (**b**) *Escherichia coli* and *Staphylococcus aureus* were treated with AgCu@MOF at concentrations ranging from 5 to 40 μg mL^−1^. (**c**) The survival of bacterial samples co-incubated with AgCu@MOF at different times, in which bacterial liquid samples were obtained every 20 min. (**d**) The bacterial samples treated with different hydrogels were divided into 500 μg mL^−1^ and 1000 μg mL^−1^ respectively. (**f**) corresponds to the survival rate of different MOFs in (**a**); (**g**) represents the inhibition rate of AgCu@MOF of different concentrations on bacterial samples; (**h**) represents the bactericidal effect of AgCu@MOF within one hour; (**i**) represents the bacteriostatic effect of PCbM hydrogels with different types and amounts of MOFs added to the hydrogels. (**e**) SEM images of normally viable, PCbM-treated, and AgCu@MOF-treated *Escherichia coli* and *Staphylococcus aureus* attached to the surface of the AgCu@MOF.

**Figure 4 polymers-15-04362-f004:**
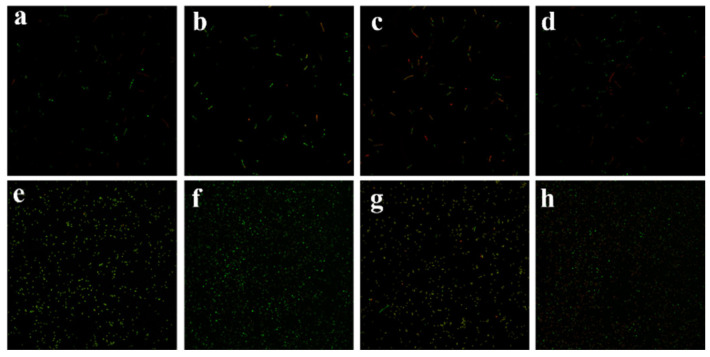
CLSM fluorescence images of (**a**) active *E. coli* and *E. coli* treated with 1 × MIC concentration of (**b**) Cu@MOF, (**c**) Ag@MOF and (**d**) CuAg@MOF for 1 h. CLSM fluorescence images of (**e**) active *S. aureus* and *S. aureus* treated with 1 × MIC concentration of (**f**) Cu@MOF, (**g**) Ag@MOF and (**h**) CuAg@MOF for 1 h.

**Figure 5 polymers-15-04362-f005:**
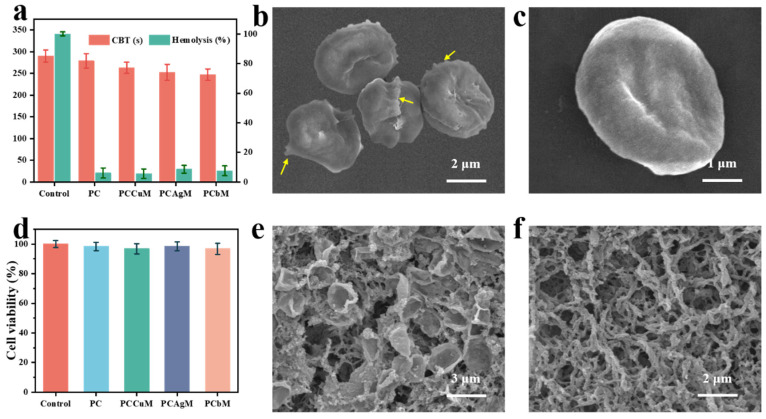
(**a**) Clotting time (**left**) and hemolysis rate (**right**) of blood samples treated with PC, PCAgM, PCCuM and PCbM hydrogels and blank control groups, respectively. SEM images of (**b**) AgCu@MOF-treated red blood cells and (**c**) normal red blood cells. (**d**) Survival rate of L929 cells co-incubated with PC, PCAgM, PCCuM and PCbM hydrogels and blank control groups. SEM images of (**e**) red blood cells and (**f**) platelets attached to the surface of PCbM hydrogel.

**Figure 6 polymers-15-04362-f006:**
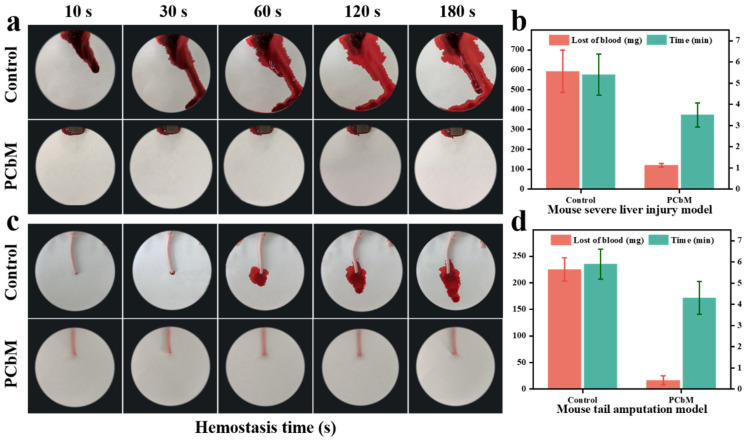
(**a**) The bleeding of the mouse liver within 10 s to 180 s after the liver was cut. The following figure shows the bleeding of the liver after applying PCbM hydrogel immediately. (**b**) The relationship between the time and amount of liver bleeding was recorded, and the PCbM experimental group was compared with the blank control group. (**c**) The bleeding situation of mice after tail amputation was compared with the bleeding situation after PCbM hydrogel enrichment. (**d**) The relationship between bleeding time and bleeding volume was recorded.

**Figure 7 polymers-15-04362-f007:**
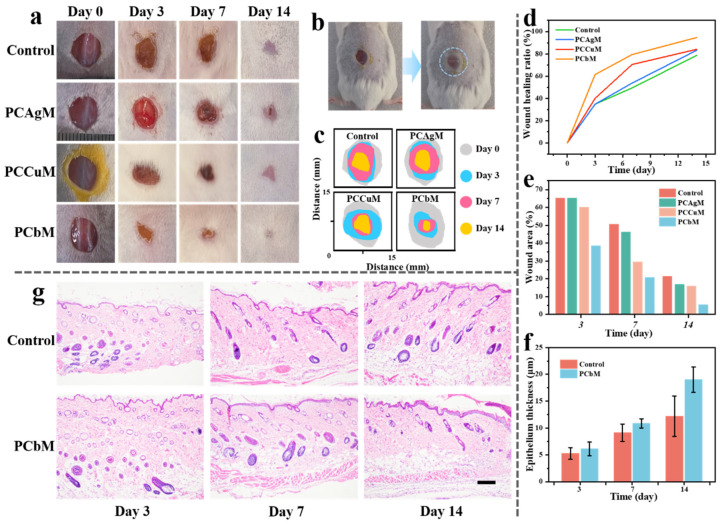
(**a**) Pictures of wounds treated with Tegaderm and PCAgM, PCCuM, and PCbM hydrogels, respectively, on days 0, 3, 7, and 14, with the diameter of the wounds on day 0 being approximately 10 mm in all cases. (**b**) Schematic representation of a hydrogel patch applied to the wound. (**c**) Relative wound area from day 0 to day 14. (**d**) Wound-healing rate from day 0 to 14. (**e**) Histogram of wound-healing area from day 0 to 14. (**f**) Quantification of the epithelial thickness. (**g**) H&E-stained images of wound pathology sections with a scale length of 100 μm.

## Data Availability

The authors confirm that the data supporting the findings of this study are available within the article.

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
