# Peer review of "Bimetal–Organic Framework-Loaded PVA/Chitosan Composite Hydrogel with Interfacial Antibacterial and Adhesive Hemostatic Features for Wound Dressings"

_polymers, 2023, doi:10.3390/polym15224362_

Round 1

Reviewer 1 Report

Dear editor,

Thanks for your invitation.

The article was checked. It needs changes before release.

-        What is the content of copper and silver in hydrogel? ICP should probably be used to determine them.

-        The method of adjusting the molar ratio of Ag and Cu copper to silver should be explained.

-        Can a structure be suggested for the synthesized compounds?

-        Is the antibacterial effect related to silver or composite? Is there a synergistic effect in this case? compared with the antibacterial properties of silver alone.

There are writing errors that need to be fixed.

Reviewer 2 Report

The present research manuscript titled “Bimetal-organic framework loaded PVA/chitosan composite hydrogel with interfacial antibacterial and adhesive hemostatic 3 features for wound dressing” by Zhang et al. is novel and very well investigated. The researchers conducted a series of in vitro and in vivo experiments to prove their claim. The results are interesting and the flow of the manuscript is good. However, the manuscript needs significant revision. My comments are as follows.

Comment 1. Abstract: Kindly include major results.

Comment 2. Most of the methods are not cited. Why?

Comment 3. References from the results section are incorrect and there is not in number.

Comment 4. The authors should state about the ANOVA and (One way or two way). Further, why did the researchers use student’s ‘t’ test? This test is not suitable for multiple comparisons. Further, the researcher should include statistical analysis in the Figures wherever required.

Comment 5. What about the stability of the hydrogels?

Comment 6. What conditions are used for SEM analyses performed? Give details.

Comment 7. Cell viability assay: The method is not clear. Further, the maintenance of cell culture and media should be discussed.

Comment 8. Metal ion release: The method is not clear. Kindly explain.

Comment 9. Overall, the language quality is poor. The scientific value of the language is poor. Kindly revise the manuscript from top to bottom. 

The language quality is poor. The scientific value of the language is poor. Kindly revise the manuscript from top to bottom. 

Reviewer 3 Report

 Zhang and the coauthors prepared a manuscript on a low-silver wound dressing material called “AgCu@MOF”, composed of a bimetallic material with silver and copper ions. This dressing exhibits strong bactericidal properties by enhancing lipophilicity and causing lipid peroxidation in bacterial membranes. Additionally, the dressing's unique properties improve water retention, tissue adhesion, and hemostatic ability, making it a promising option for wound treatment. However, the identity of the “AgCu@MOF” was not well characterized which makes this manuscript problematic. This manuscript cannot be published without knowing the exact structure of “Cu@MOF”, “Ag@MOF” and “CuAg@MOF”.

Comments:

1.       The authors claimed that they have synthesized Ag/Cu contained MOFs. However, the structural characterizations of the “MOF” is not sufficient. What is the crystal structure of the synthesized “MOFs”? What is the topology of the “MOFs”? Without knowing the exact structure, it is inappropriate to call the synthesized material “MOFs”. It could be just one-dimensional coordination polymer or discrete metal complexes. Without knowing the structural identities of the “Cu@MOF”, “Ag@MOF” and “CuAg@MOF”, this manuscript cannot be published.

2.       MOFs should have intrinsic porosity. What is the porosity of the “Cu@MOF”, “Ag@MOF” and “CuAg@MOF” compounds?

3.       In section 3 references errors were found. Please fix!

4.       The authors performed leaching test on the “Cu@MOF”, “Ag@MOF” and “CuAg@MOF”. What is the water stability of these compounds? Was the crystallinity maintained after being treated with water?

Round 2

Reviewer 2 Report

The revision is satisfactory. I don't have further comments.

Reviewer 3 Report

Comments have been addressed.